# Effect of Ethylene Oxide Exposure on Sleep Health: Using NHANES Data from 2015 to 2020

**DOI:** 10.3390/healthcare12242499

**Published:** 2024-12-11

**Authors:** Hansol Choi, Yoon-Soo Choy

**Affiliations:** 1Department of Preventive Medicine, Yonsei University College of Medicine, 50-1 Yonsei-ro, Seodaemun-gu, Seoul 03722, Republic of Korea; hansolchoi@yuhs.ac; 2Department of Smart Healthcare Information, Healthcare Management, Eulji University, 553 Sanseong-daero, Sujeong-gu, Seongnam-si 13135, Republic of Korea

**Keywords:** chemical exposure, hemoglobin adducts of ethylene oxide (HbEO), sleep duration, sleep quality, sleep health

## Abstract

**Background/Objectives**: This study aims to investigate the effects of ethylene oxide (EO) exposure on sleep health, focusing on sleep duration and quality. **Methods**: The study analyzed data from the NHANES (National Health and Nutrition Examination Survey) 2015–2020 cycles, including 4268 participants aged 20 and older. EO exposure was measured using hemoglobin adducts of EO (HbEO), which serve as a reliable biomarker. Sleep health was assessed through self-reported questionnaires on sleep duration and quality. Participants were categorized based on sleep duration (<6 h, 6–9 h, >9 h) and symptoms of sleep disturbances. Statistical analyses employed survey-weighted logistic regression models to evaluate the associations between HbEO levels and sleep outcomes, adjusting for sociodemographic, health-related, and behavioral factors. Moreover, to examine whether the impact of ethylene oxide exposure on sleep quality and sleep duration varies by sociodemographic characteristics, stratified analyses were conducted based on gender, age, marital status, and employment type. **Results**: According to the results, higher EO exposure was associated with shorter sleep durations and increased likelihood of sleep disturbances. Moreover, according to sub-group analysis by sex, men with higher exposure to EO, were likely to have short sleep duration, and women with higher exposure to EO had higher risk of daytime sleepiness and sleep problems. **Conclusions**: The findings suggest that EO exposure may negatively impact sleep health, emphasizing the need for stricter EO exposure regulations and public health interventions to reduce associated risks.

## 1. Introduction

Sleep is an essential component of human health and well-being, playing a critical role in maintaining physiological, psychological, and cognitive functions [1]. The quality and quantity of sleep can be influenced by a variety of environmental, physiological, and chemical factors [2]. In addition, adequate sleep duration must be ensured to maintain homeostasis, cognitive function, and overall health [3].

Sleep duration and quality are essential determinants of health, influencing physical, mental, and cognitive well-being. In the United States, approximately one-third of adults fail to achieve the recommended seven hours of sleep per night, as reported by a previous study [4]. Additionally, obstructive sleep apnea (OSA), a common sleep disorder, affects nearly 30 million Americans, with around 80% of cases remaining undiagnosed [5]. These patterns highlight the widespread prevalence of sleep-related health issues and their potential impact on public health.

Both insufficient and excessive sleep durations have been linked to adverse health outcomes. A previous meta-analysis showed that short sleep durations, typically less than six hours per night, are associated with an increased risk of cardiovascular diseases, type 2 diabetes, and all-cause mortality [6]. Conversely, another study demonstrated that long sleep durations exceeding nine hours are correlated with higher risks of obesity, cognitive decline, and reduced overall health outcomes [7]. These findings emphasize that deviations from the recommended sleep duration, in either direction, can have detrimental effects on health.

In addition to sleep duration, sleep quality plays a critical role in overall well-being. Poor sleep quality, characterized by fragmented or non-restorative sleep, is strongly associated with mental health challenges such as depression and anxiety, as highlighted in [8]. Moreover, disrupted sleep has been shown to elevate inflammatory markers, increasing susceptibility to chronic illnesses, including cardiovascular diseases and diabetes [9].

Disturbances in sleep patterns or sleep quality have been associated with a myriad of health problems, including metabolic disorders, cardiovascular diseases, impaired immune function, and neurodegenerative conditions [10]. There is a growing body of research linking environmental exposures to sleep disturbances, thus exacerbating health risks for exposed populations [2,11,12]. Recent studies have increasingly focused on the impact of various chemicals on sleep, with EO being one such substance of interest [13,14,15,16]

EO is a chemical primarily used in industrial applications, including as a sterilizing agent, a chemical intermediate, and in pharmaceutical manufacturing [17]. EO is a colorless, flammable gas with a slightly sweet odor, commonly used in the sterilization of medical equipment and the production of chemicals like ethylene glycol, which is a key ingredient in antifreeze and polyester [18,19]. Due to its effectiveness in eliminating bacteria, viruses, and fungi, EO is widely employed in various industrial and healthcare settings.

Despite its utility, EO poses significant risks to human health, particularly due to its classification as a Group 1 carcinogen by the International Agency for Research on Cancer (IARC) [13]. Prolonged exposure to EO has been linked to an increased risk of cancers, particularly hematopoietic cancers such as leukemia and lymphoma, as well as reproductive and developmental effects [20,21]. Moreover, recent research has raised concerns about the potential effects of EO exposure on the nervous system, suggesting that it could directly impact sleep patterns and quality [16,22,23].

To understand the effects of EO on sleep, it is essential to first explore how this chemical interacts within the human body [1]. EO undergoes various biochemical pathways during metabolism, and its metabolites may affect the nervous system [24]. Notably, the possibility that EO could alter neurotransmitter balance or exert direct neurotoxic effects presents a significant avenue for investigating its impact on sleep [25]. While the carcinogenic and mutagenic properties of EO have been extensively studied, its potential impact on sleep has not received the same level of attention [26]

The mechanisms through which EO might impact sleep are not fully understood, but it is hypothesized that EO’s ability to induce oxidative stress and DNA damage could play a significant role [25,26]. Oxidative stress has been implicated in the pathogenesis of various sleep disorders, including insomnia and sleep apnea [2]. Moreover, EO’s neurotoxic effects, potentially mediated through disruptions in neurotransmitter systems and neuroinflammation, could further contribute to alterations in sleep architecture [27,28]

Current research on EO has predominantly focused on its toxicity and carcinogenicity, with relatively few studies directly examining its effects on sleep [29]. This study seeks to fill the gap in the literature by systematically investigating the effects of EO exposure on sleep duration [30]. Specifically, this research explores the relationship between different levels and durations of EO exposure and changes in sleep patterns. Additionally, the study assesses the correlation between EO exposure and sleep-related biomarkers, such as cortisol levels, melatonin secretion, and inflammatory markers, which are known to influence sleep regulation. By doing so, it seeks to contribute valuable insights into the potential health risks associated with EO exposure and inform the development of safety guidelines for its use.

## 2. Materials and Methods

### 2.1. Study Population

We used data from the NHANES (National Health and Nutrition Examination Survey), which is an ongoing program of studies designed to evaluate the health and nutritional status of adults and children in the United States. A detailed description of the survey protocol has been reported elsewhere [31,32]. In this analysis, we included the data from two cycles of the NHANES through the 2015–2016 and 2017–2020 surveys to analyze the association between EO exposure and sleep health. As shown in Figure 1, after the exclusion of 20,763 participants with insufficient data regarding exposure and outcome and 520 participants who were <20 years old, a total of 4268 participants were included in the current analysis. All procedures were approved by the National Center for Health Statistics (NCHS) Ethics Review Board (https://www.cdc.gov/nchs/nhanes/irba98.htm, accessed on 6 April 2024.), and all participants provided written informed consent.

### 2.2. Measurements

Since EO has a half-life of about 42 min in humans, and the use of hemoglobin adducts of ethylene oxide (HbEO) has been demonstrated to be a reliable and sensitive method for measuring exposure, we used HbEO, which has a biological half-life of 4 months. The exposure of HbEO was measured according to the description in the NHANES laboratory procedure manual [33].

Sleep health was measured by a self-reported questionnaire, assessing both the quantity and quality of sleep. Sleep duration was evaluated by asking “How much sleep usually get at night on weekdays or workdays?”. In the present study, participants were divided into three groups based on their reported sleep duration: <7 h/day, “insufficient sleep group”; 7 to <9 h/day, “sufficient sleep group”; and ≥9 h/day, “excessive sleep group” according to the National Sleep Foundation’s updated sleep duration recommendations [34]. In addition to these criteria, the “very insufficient sleep group” was added to further assess participants who reported sleeping <6 h/day, to evaluate the impact of HbEO on extremely short sleep duration.

Sleep quality was evaluated based on several factors, including the frequency of snoring and snorting, gasping, or stopping breathing per week, the presence of day sleepiness, sleep problems, and obstructive sleep apnea in the last 12 months. Snoring was defined as occurring ≥1/week, and snorting, gasping, or stopping breathing was similarly defined as occurring ≥1/week. Day sleepiness was assessed by asking how often participants felt excessively sleepy or overly sleepy during the day in the past month, with ≥5/month indicating day sleepiness. Sleep problems were defined as any of the following: doctor diagnosis of trouble sleeping; or trouble falling asleep, staying asleep, or sleeping too much ≥several days/2 weeks [35]. Obstructive sleep apnea was defined as any of the following: snoring ≥3/week; or snorting, gasping, or stopping breathing ≥3/week; or day sleepiness ≥16–30/month despite sleeping ≥7 h/day [36].

As we used self-reported questionnaire data from the NHANES (National Health and Nutrition Examination Survey), our alpha values are influenced by the limited number of items and the exploratory nature of our instrument. Research indicates that with fewer items, Cronbach’s alpha tends to be lower, yet the instrument can still maintain validity [37]. Similarly, another study emphasizes that alpha should not be the sole indicator of internal consistency, especially in exploratory research with fewer items [38]. Considering these insights, we have clarified our methodological approach to reflect the exploratory context and the constraints related to the number of items in our study.

### 2.3. Statistical Analysis

Continuous variables with normal distribution were reported as weighted mean with standard errors and categorical variables were expressed as numbers with weighted percentages. Survey-weighted logistic regression models were used to assess the independent associations of HbEO with sleep health. Sociodemographic variables (age, sex, race, economic status, marital status, and job status), health-related variables (body mass index, history of cardiovascular disease and cancer, hypertension, diabetes mellitus, and depressive symptoms), and behavioral variables (physical activity, sedentary activity, caffeine consumption, alcohol drinking, and cotinine) were entered into the model as covariates. Furthermore, to examine whether the impact of EO exposure on sleep quality and sleep duration varies by sociodemographic characteristics, stratified analyses were conducted based on gender, age, marital status, and employment type. All statistical analyses were conducted using SAS software (version 9.4 SAS, Cary, NC, USA) and SPSS 23. Statistical significance was defined as a two-sided *p*-value less than 0.05.

## 3. Results

This study investigated the impact of EO exposure on sleep duration and quality. A total of 4248 participants with an average age of 48.4 years were included in the analysis. The results of the participants’ sleep quality, sleep duration, and levels of EO exposure are presented in Table 1. According to the analysis of EO exposure based on sleep duration, individuals who slept fewer than 6 h were exposed to a high level of ethylene oxide, averaging 99.3 pmol/g Hb. Additionally, individuals experiencing sleep problems such as snoring, sleep apnea, and daytime sleepiness were more likely to be exposed to higher levels of ethylene oxide.

Regarding sociodemographic characteristics, analysis of EO exposure by age revealed that adults in their 50s had the highest exposure level at 69.4 pmol. Furthermore, Black individuals, those with lower education levels, and those with lower income levels were relatively more exposed to ethylene oxide. In terms of health behavior characteristics, individuals who consumed high amounts of caffeine, drank alcohol frequently, or were current smokers were also more likely to be exposed to higher levels of ethylene oxide.

After adjusting for other control variables, the analysis of the effect of EO exposure on sleep duration (Table 2) showed a trend where higher EO exposure was associated with shorter sleep duration, but this was not statistically significant. When analyzed by gender, no significant effect was observed in women. However, among men, those in the highest exposure group had significantly shorter sleep durations compared to those in the lowest exposure group (β = −0.0328, *p*-value < 0.05).

The analysis of the impact of EO exposure on sleep duration and quality (Table 3) revealed that individuals with higher exposure levels had a statistically significant higher likelihood of sleeping for fewer than 6 h (OR = 1.205, 95% CI 1.049–1.384). However, there was no observed impact of EO exposure on overall sleep quality. Nevertheless, when EO exposure was categorized into quartiles, individuals with higher exposure levels were significantly more likely to have shorter sleep durations (<6 h) compared to those with the lowest exposure levels. In terms of sleep quality, individuals with higher EO exposure levels were more likely to experience daytime sleepiness (OR = 1.441, 95% CI 1.001–2.075) and an increased likelihood of sleep problems (OR = 1.415, 95% CI 1.019–1.965) compared to those with lower exposure.

A stratified analysis was conducted to evaluate the impact of EO exposure on sleep health across socioeconomic characteristics, such as sex (Appendix A), age (Appendix A), marital status (Appendix A), and job status (Appendix A). The stratified analysis demonstrated that higher ethylene oxide (EO) exposure negatively impacts sleep health across various socioeconomic groups. Men exposed to higher EO levels were 1.173 times more likely to sleep fewer than 6 h (OR = 1.173, 95% CI 1.009–1.365) and had significantly shorter sleep durations (OR = 2.589, 95% CI 1.285–5.216), while women showed similar patterns with an OR of 1.386 (95% CI 1.073–1.788) and increased daytime fatigue (OR = 1.697, 95% CI 1.059–2.719). Among individuals under 40, short sleep duration (<6 h) was 1.458 times more likely (OR = 1.458, 95% CI 1.319–1.800), while those in their 40s experienced greater daytime sleepiness (OR = 1.516, 95% CI 1.138–1.943). Older adults (60+) showed higher risks for sleep apnea (OR = 1.234, 95% CI 1.061–1.888) and obstructive sleep apnea (OR = 1.306, 95% CI 1.107–1.875), with individuals aged 70+ having 2.153 times the likelihood of short sleep duration (OR = 2.153, 95% CI 1.482–9.616). Married individuals with high EO exposure were more likely to experience short sleep (<6 h) (OR = 1.304, 95% CI 1.089–1.560) and breathing-related disturbances (OR = 1.346, 95% CI 1.121–1.616), while unmarried individuals were 3.013 times more likely to report severe sleep problems (OR = 3.013, 95% CI 1.101–8.247). Employed individuals with higher EO exposure faced increased sleep problems (OR = 1.742, 95% CI 1.075–3.113), while unemployed individuals were more likely to experience short sleep (OR = 1.270, 95% CI 1.043–1.546) and breathing-related issues (OR = 1.209, 95% CI 1.097–1.466). These findings highlight the consistent and statistically significant negative effects of higher EO exposure on sleep duration and quality across diverse groups.

These findings suggest that higher ethylene oxide (EO) exposure is significantly associated with increased risks of short sleep duration, poor sleep quality, and breathing-related sleep disturbances across various demographic and socioeconomic groups, with specific patterns varying by sex, age, marital, and employment status.

## 4. Discussion

This study analyzed the impact of EO exposure on sleep duration and sleep quality using NHANES data. The concentration of hemoglobin adducts of EO in the blood was used as an indicator of exposure, and variables such as self-reported sleep duration, snoring, and daytime sleepiness were used to assess sleep quality and duration. First, multiple linear regression analysis was conducted to determine the effect of EO exposure on sleep duration according to gender. The level of EO exposure was analyzed as both a continuous and categorical variable. Additionally, to test the hypotheses that higher EO exposure would lead to shorter sleep duration and negatively affect sleep quality—such as increased snoring, sleep apnea, and daytime sleepiness—survey-weighted logistic regression was performed.

Although studies using NHANES data to analyze the effects of EO exposure on sleep duration and quality through survey-weighted logistic regression and multiple linear regression are rare, there have been studies that analyzed the effects of environmental pollutants on sleep using NHANES data.

A previous study investigated the impact of polycyclic aromatic hydrocarbon (PAH) exposure on sleep quality and duration using NHANES data among adults aged 20 and older, with urinary metabolite concentrations as the exposure indicator. The authors utilized survey-weighted logistic regression and multiple linear regression for their analysis. The results showed a significant association between high PAH exposure and reduced sleep duration, as well as an increased risk of sleep apnea [39].

Additionally, another study explored the effects of heavy metals such as cadmium (Cd), lead (Pb), and mercury (Hg) on sleep duration and sleep apnea using NHANES data for adults aged 20 and older [40]. The authors used survey-weighted logistic regression to analyze the effects of metal concentrations on sleep disorders and multiple linear regression to assess the impact of metal levels on sleep duration. The results indicated that higher exposure to heavy metals, especially cadmium and lead, was associated with an increased likelihood of short sleep duration (fewer than 6 h) and that lead and mercury exposures were linked to a higher risk of sleep apnea. In addition, there have been studies analyzing the association between volatile organic compounds (VOCs) and decreased sleep quality.

Moreover, the relationship between HbEO and short sleep duration among 3438 adults aged 20 years and older are analyzed utilizing data from the National Health and Nutrition Examination Survey (NHANES) [25], and other studies have utilized NHANES data similar to this research to examine the impact of EO exposure on cardiovascular health metrics [41]

The findings of the present study on the effects of EO exposure on sleep quality and duration highlight a significant public health concern. The study reveals that individuals exposed to EO tend to experience markedly reduced sleep quality and shorter sleep durations. These results suggest that EO may disrupt sleep through its impact on the nervous and hormonal systems, which could have severe consequences for overall health. This study’s importance is underscored when compared to the existing literature, which has extensively documented the adverse health effects of ethylene oxide.

Prior research has consistently shown that exposure to EO is linked to various health issues, including cancer, neurological disorders, and respiratory problems. For example, Steenland et al. conducted a comprehensive study involving workers with long-term exposure to EO and found a significant increase in cancer incidence and mortality rates [42]. This study highlighted the critical health risks associated with ethylene oxide, reinforcing the notion that such exposure could have broad-ranging effects, including the deterioration of sleep quality due to the substance’s impact on the nervous and immune systems.

Similarly, A previous meta-analysis showed a strong association between EO exposure and various cancers, particularly hematologic cancers, and breast cancer. Chronic health issues like these can lead to psychological stress and physical discomfort, both of which are well-known factors that contribute to poor sleep quality [43]. Moreover, other previous research found that women exposed to EO exhibited a higher incidence of breast cancer, a condition often linked to hormonal imbalances. Hormonal disruptions are critical in maintaining circadian rhythms and sleep cycles, suggesting that EO exposure could directly interfere with sleep [44].

In addition, EO exposure can rapidly increase EO levels in the blood, potentially leading to immediate physiological responses such as sleep disturbances. Continuous exposure to such acute changes could result in long-term reductions in sleep quality and duration, severely affecting daily functioning and overall health [45].

Moreover, as indicated by earlier studies on the effect of exposure to other harmful chemicals, such as formaldehyde, on respiratory health, particularly in children [46], exposure to such chemicals could increase the likelihood of developing asthma, which is closely linked to sleep disturbances. This study underscores the importance of investigating the effects of harmful chemicals like EO on sleep, advocating for more research in this area.

Another study assessed the health risks associated with EO exposure in workers and reported a range of adverse health outcomes, including neurotoxicity and reproductive issues [47]. These health problems are often accompanied by sleep disturbances, indicating a possible link between EO exposure and impaired sleep.

Following this latter study, a 15-year follow-up study was carried out, investigating workers in chemical manufacturing industries exposed to EO, which confirmed the elevated risk of mortality from cancer and other diseases. The long-term nature of these health effects further emphasizes the need to understand how chronic exposure to EO might impact sleep over time [48].

In line with findings from other studies that focused on the quantitative assessment of EO exposure and its impact on worker health, this study provided critical insights into the levels of exposure that could potentially lead to significant health problems, including sleep-related issues. The authors’ work suggests that even low levels of EO exposure might have measurable effects on health, particularly in vulnerable populations such as those with existing sleep disorders.

According to the results of previous research, there is a significant association between higher HbEO levels and an increased risk of short sleep duration (defined as ≤6 h per night) [22], and increased risk of depressive symptoms among the study population [25]. Moreover, elevated HbEO levels were linked to lower overall cardiovascular health scores, with more pronounced associations observed above certain exposure thresholds [41].

Given this extensive background, it is clear that in-depth studies on the effects of EO on sleep quality and duration are not only warranted but urgently needed. Sleep is a fundamental process for both physical and mental recovery, and disruptions in sleep can lead to numerous adverse outcomes, including fatigue, impaired immune function, and mental health disorders. The risks posed by exposure to industrial chemicals like EO extend beyond occupational settings, potentially affecting the general public as well.

This study contributes to a deeper understanding of how EO affects sleep and lays the groundwork for developing practical prevention and management strategies. Effective regulation of EO exposure is crucial, as is the early identification and management of sleep problems in individuals exposed to this chemical. In addition, this study distinguishes itself from previous research by analyzing the impact of EO exposure on sleep quality and duration with a specific focus on gender differences. While prior studies primarily concentrated on the general health risks associated with EO, our research separately examines the effects on sleep health for both men and women, thereby identifying gender-specific impacts of EO exposure. This approach highlights the possibility that EO exposure may affect individuals differently based on gender, providing crucial evidence to support gender-specific public health policies and exposure standards aimed at safeguarding sleep health. Future research should aim to elucidate the precise mechanisms by which EO impacts sleep and explore various approaches to mitigate these effects. Such research will be vital in protecting both workers and the broader population from the harmful consequences of EO exposure.

Furthermore, this study proposes actionable suggestions to improve the reliability and validity of the Sleep Pattern Questionnaire (SPQ) for future research. First, integrating objective sleep data, such as those collected from wearable devices like smartwatches, can enhance the accuracy of sleep measurements by complementing self-reported responses. Second, incorporating additional variables, such as daytime activity levels and stress assessments, into the SPQ can provide a more comprehensive understanding of the factors influencing sleep patterns. These adaptations have the potential to refine the precision and applicability of the SPQ, making it a more robust tool for future studies. We strongly recommend adopting these enhancements to strengthen the reliability and validity of the SPQ in subsequent research efforts.

However, this study has several limitations. First, the measurement of sleep quality and duration was based on self-reported surveys. Since self-reported surveys were used, participants might not have accurately remembered or reported their sleep patterns or other relevant health information, which can introduce bias. More objective measurements, such as polysomnography, would be ideal for accurately assessing sleep quality and duration. Since the data were gathered from individuals’ experiences, it is difficult to precisely measure sleep disorders like snoring and daytime sleepiness.

Furthermore, the study used a cross-sectional design to analyze the impact of exposure on sleep quality and duration, which limits the ability to capture changes in exposure over time. As a result, establishing causal relationships is challenging. However, since the study utilized NHANES data, the sample size was large enough to analyze the association between the two variables effectively.

Moreover, the study may not have had granular data on the exact duration or intensity of exposure, which could limit the ability to fully assess the relationship between EO exposure and sleep outcomes. To understand the association between EO exposure and sleep outcomes, more detailed data on the duration and intensity of EO exposure might be needed.

Another limitation is that certain factors directly affecting sleep quality, such as participants’ physical and psychological health, and lifestyle factors like diet, physical activity, or other environmental pollutants, were not controlled. Additionally, the study did not account for exposure to other compounds that may influence sleep.

However, this study holds significant importance from both policy and epidemiological perspectives. From a policy standpoint, it emphasizes the potential health risks associated with EO exposure, particularly regarding sleep quality. The findings suggest that stricter regulations on industrial emissions of EO are necessary to protect public health, especially in communities located near industrial facilities. This research could help inform policies that regulate EO emissions and ensure that occupational safety standards are strengthened in industries where EO is commonly used, such as sterilization plants. Workers in these environments would benefit from enhanced protective measures, including regular monitoring of EO levels to reduce their long-term health risks. Furthermore, public health campaigns could raise awareness about the potential hazards of EO exposure, not only among industrial workers but also among the general public, especially those living in high-exposure areas.

From an epidemiological perspective, this study contributes to a broader understanding of how environmental pollutants like EO affect public health, specifically by linking such exposure to sleep disturbances. Sleep disorders are recognized precursors to more severe health conditions, such as cardiovascular disease and diabetes. By demonstrating the connection between EO exposure and poor sleep quality, this research highlights the need for more comprehensive health risk assessments and surveillance programs, particularly in high-exposure populations. Additionally, it underscores the necessity of longitudinal studies to better understand the causal relationship between prolonged EO exposure and sleep disturbances over time. Such studies could lead to more effective monitoring and targeted interventions to address the long-term health impacts of environmental pollutants.

## 5. Conclusions

This study highlights a significant association between EO exposure and impaired sleep health, as higher levels of EO exposure are linked to shorter sleep durations and increased sleep disturbances. Moreover, according to stratified analyses based on sociodemographic characteristics, higher exposure to EO has statistically significant negative effects on both sleep duration and quality. These findings underscore the potential impact of EO on the nervous and hormonal systems, which may disrupt sleep regulation. Given the critical role of sleep in overall health, the results advocate for stricter regulatory standards on EO exposure, especially in industrial and healthcare settings. This research supports the need for public health interventions aimed at monitoring and mitigating EO exposure risks to promote better sleep health and reduce long-term adverse health outcomes.

## Figures and Tables

**Figure 1 healthcare-12-02499-f001:**
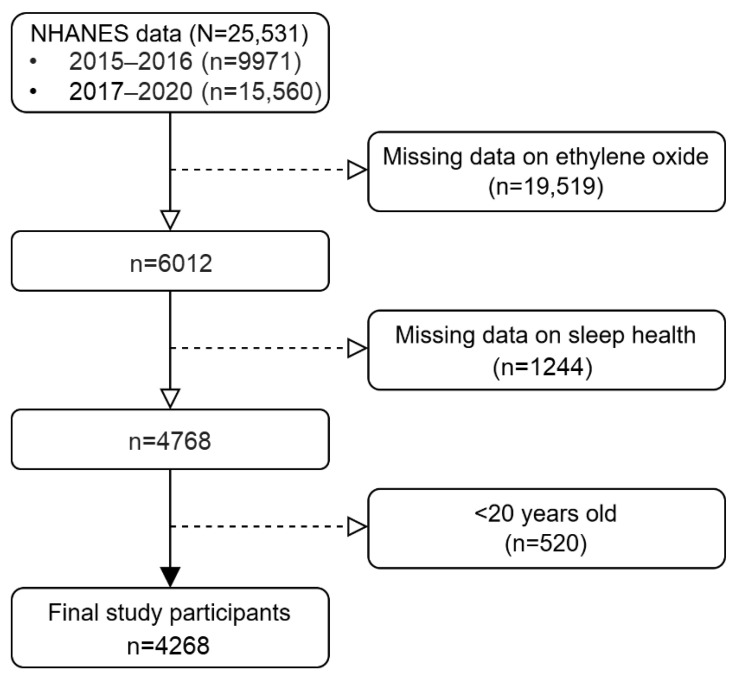
Flow chart of the study participants.

**Table 1 healthcare-12-02499-t001:** General characteristics of study participants according to exposure to EO (*n* = 4248).

Variables	Total (*n* = 4248)	EO, pmol/g Hb	Men (*n* = 2096)	Women (*n* = 2152)	*p*
Age, years old	48.4 ± 0.6		47.8 ± 0.7	48.9 ± 0.7	0.115
<40	1330 (31.3)	61.4 ± 5.2	631 (30.1)	699 (32.5)	0.106
40–49	672 (15.8)	59.0 ± 6.5	310 (14.8)	362 (16.8)	
50–59	730 (17.2)	69.4 ± 8.2	360 (17.2)	370 (17.2)	
60–69	817 (19.2)	50.7 ± 4.2	427 (20.4)	390 (18.1)	
≥70	699 (16.5)	34.6 ± 3.3	368 (17.6)	331 (15.4)	
Race, %					
Non-Hispanic White	1440 (33.9)	55.7 ± 4.2	736 (35.1)	704 (32.7)	0.286
Non-Hispanic Black	1014 (23.9)	84.5 ± 5.1	504 (24.0)	510 (23.7)	
Mexican American/Hispanic	1080 (25.4)	41.1 ± 3.0	512 (24.4)	568 (26.4)	
Non-Hispanic Asian/Others	714 (16.8)	62.0 ± 7.0	344 (16.4)	370 (17.2)	
Education level, % (n = 4244)					
<High school	864 (20.4)	85.5 ± 6.0	456 (21.8)	408 (19.0)	0.247
High school or equivalent	1031 (24.3)	78.6 ± 8.0	526 (25.1)	505 (23.5)	
Some college or Associate’s degree	1288 (30.3)	56.8 ± 4.0	582 (27.8)	706 (32.8)	
≥Bachelor’s degree	1061 (25.0)	29.5 ± 1.6	528 (25.2)	533 (24.8)	
Marital status, % (n = 4245)					
Married/Cohabiting	2576 (60.7)	50.5 ± 3.1	1373 (65.5)	1203 (56.0)	<0.0001
Divorced/Widowed/Separated	901 (21.2)	78.0 ± 9.0	332 (15.8)	569 (26.5)	
Never married	768 (18.1)	61.1 ± 4.9	390 (18.6)	378 (17.6)	
Family income to poverty ratio, % (n = 3715)					
≤1.30	1072 (28.9)	86.8 ± 6.9	494 (27.1)	578 (30.5)	0.082
1.30–1.85	536 (14.4)	84.4 ± 7.5	263 (14.4)	273 (14.4)	
1.85–4.00	1156 (31.1)	54.5 ± 3.2	579 (31.8)	577 (30.5)	
>4.00	951 (25.6)	37.1 ± 4.5	485 (26.6)	466 (24.6)	
Job, % (n = 4243)					
With a job or business	2444 (57.6)	52.1 ± 3.5	1301 (62.1)	1143 (53.2)	<0.0001
Looking for work/Not working	1799 (42.3)	66.6 ± 3.8	794 (37.9)	1005 (46.8)	
EO, pmol/g Hb	57.2 ± 3.0		62.1 ± 3.3	52.7 ± 3.9	0.027
Cotinine, ng/mL (n = 4167)	52.6 ± 3.8		69.7 ± 5.5	36.3 ± 3.8	<0.0001
hs-CRP, mg/L (n = 4132)	3.8 ± 0.2		3.1 ± 0.1	4.4 ± 0.2	<0.0001
Sleep duration, hours/day (last 12 months)	7.6 ± 0.0		7.4 ± 0.0	7.8 ± 0.0	<0.0001
<7, insufficient	1058 (24.9)	74.9 ± 7.2	591 (28.2)	467 (21.7)	<0.0001
7–9, moderate	2219 (52.2)	48.1 ± 2.5	1098 (52.4)	1121 (52.1)	
≥9, excessive	971 (22.9)	63.1 ± 5.5	407 (19.4)	564 (26.2)	
<6, very insufficient	399 (9.4)	99.3 ± 18.0	232 (11.1)	167 (7.8)	<0.0001
Sleep quality (last 12 months)					
Snoring, ≥1/week	2913 (68.6)	55.6 ± 3.1	1533 (73.1)	1380 (64.1)	<0.0001
Snorting, gasping, or stopping breathing, ≥1/week	1003 (23.6)	69.3 ± 5.5	594 (28.3)	409 (19.0)	<0.0001
Day sleepiness	1091 (25.7)	64.1 ± 4.9	498 (23.8)	593 (27.6)	0.013
Sleep problems	1989 (46.8)	63.0 ± 4.4	908 (43.3)	1081 (50.2)	<0.0001
Obstructive sleep apnea	2057 (48.4)	58.0 ± 3.9	1105 (52.7)	952 (44.2)	<0.0001
Caffeine intake, mg/d (last 30 days) (n = 3897)	180.9 ± 7.0		208.8 ± 9.3	154.6 ± 7.8	<0.0001
<400, within the recommended range	3648 (93.6)	52.0 ± 2.8	1774 (91.7)	1874 (95.5)	0.001
≥400, exceeding the recommended range	249 (6.4)	109.2 ± 11.8	161 (8.3)	88 (4.5)	
Alcohol drinking, % (past 12 months) (n = 2213)					
Never drank	473 (21.4)	71.1 ± 10.0	240 (21.2)	233 (21.6)	<0.0001
<1/week	1084 (49.0)	59.3 ± 4.8	468 (41.3)	616 (57.0)	
1–4/week	499 (22.5)	57.0 ± 5.8	305 (26.9)	194 (17.9)	
>4/week	157 (7.1)	56.8 ± 7.1	119 (10.5)	38 (3.5)	
Habitual drinker, ≥1/week	656 (15.4)	57.0 ± 5.2	424 (20.2)	232 (10.8)	<0.0001
Cigarette smoking, % (n = 4247)					
Never smoked	2414 (56.8)	23.9 ± 1.0	985 (47.0)	1429 (66.4)	<0.0001
Former smoker	1027 (24.2)	32.2 ± 1.3	630 (30.1)	397 (18.4)	
Current smoker	806 (19.0)	206.7 ± 8.2	480 (22.9)	326 (15.1)	
Body mass index, kg/m^2^	29.6 ± 0.2		29.3 ± 0.2	30.0 ± 0.3	0.027
<18.5, Underweight	58 (1.4)	161.6 ± 37.8	26 (1.2)	32 (1.5)	<0.0001
18.5–24.9, Normal	1062 (25.0)	68.8 ± 4.8	503 (24.0)	559 (26.0)	
25.0–29.9, Overweight	1388 (32.7)	52.7 ± 3.6	793 (37.8)	595 (27.6)	
≥30, Obese	1740 (41.0)	51.0 ± 4.6	774 (36.9)	966 (44.9)	
History of diseases, %					
Cardiovascular disease	300 (7.1)	66.0 ± 10.1	183 (8.7)	117 (5.4)	0.199
Cancer	269 (6.3)	47.9 ± 5.2	131 (6.3)	138 (6.4)	0.118
Current status of diseases, %					
Hypertension					
Diabetes mellitus					
PHQ-9 score, 0–27	2.9 ± 0.1		2.5 ± 0.1	3.2 ± 0.1	<0.0001
≥10, depressive symptom	320 (7.5)	107.4 ± 11.5	130 (6.2)	190 (8.8)	0.066

Data expressed as weighted mean ± standard error or number (percentage). EO: Ethylene Oxide, PHQ-9: Patient Health Questionnaire-9.

**Table 2 healthcare-12-02499-t002:** Association between EO concentration and sleep duration according to sex.

Variables	Unadjusted β	95% CI	*p*	* Adjusted β	95% CI	*p*
EO, pmol/g Hb						
Total						
Continuous, ln	−0.039	(−0.084, −0.005)	0.084	−0.031	(−0.095, −0.034)	0.338
Q1	Ref			Ref		
Q2	−0.008	(−0.164, −0.147)	0.916	−0.047	(−0.223, −0.130)	0.596
Q3	−0.086	(−0.249, −0.076)	0.288	−0.102	(−0.270, −0.065)	0.224
Q4	−0.094	(−0.273, −0.085)	0.296	−0.076	(−0.323, −0.172)	0.541
Men						
Continuous, ln	−0.055	(−0.108, −0.002)	0.041	−0.068	(−0.144, −0.008)	0.078
Q1, ≤16.7	Ref			Ref		
Q2, 16.7 > –23.2	−0.038	(−0.259, −0.183)	0.730	−0.115	(−0.365, −0.134)	0.356
Q3, 23.2 > –71.5	−0.113	(−0.358, −0.132)	0.357	−0.182	(−0.449, −0.084)	0.175
Q4, ≥71.5	−0.243	(−0.491, −0.005)	0.055	−0.328	(−0.635, −0.021)	0.037
Women						
Continuous, ln	0.007	(−0.059, −0.072)	0.842	0.004	(−0.081, −0.088)	0.930
Q1, ≤15.8	Ref			Ref		
Q2, 15.8 > –21.4	0.000	(−0.195, −0.195)	0.997	0.026	(−0.191, −0.243)	0.813
Q3, 21.4 > –34.3	−0.042	(−0.240, −0.155)	0.666	−0.039	(−0.263, −0.185)	0.727
Q4, ≥34.3	0.042	(−0.176, −0.259)	0.701	0.159	(−0.131, −0.450)	0.275

* Adjusted for sociodemographic variables (age, sex, race, economic status, marital status, and job status), health-related variables (body mass index, history of cardiovascular disease and cancer, hypertension, diabetes mellitus, and depressive symptom), and behavioral variables (physical activity, sedentary activity, caffeine consumption, alcohol drinking, and cotinine). EO: Ethylene Oxide, CI: Confidence Intervals.

**Table 3 healthcare-12-02499-t003:** Adjusted odds ratios for sleep health according to EO concentration.

Variables	* Adjusted OR (95% CI) for Sleep Health								
	Continuous, ln		Q1	Q2		Q3		Q4	
Sleep duration, hours/day (last 12 months)									
<6, very insufficient	1.205	(1.049, 1.384)	Ref	2.118	(1.147, 3.912)	2.026	(1.201, 3.418)	2.088	(1.301, 3.350)
<7, insufficient	1.107	(1.001, 1.224)	Ref	1.062	(0.714, 1.579)	1.105	(0.768, 1.592)	1.418	(0.972, 2.069)
≥9, excessive	1.022	(0.873, 1.197)	Ref	1.170	(0.815, 1.679)	1.028	(0.700, 1.511)	1.159	(0.666, 2.018)
Sleep quality (last 12 months)									
Snoring, ≥1/week	1.010	(0.917, 1.113)	Ref	0.861	(0.643, 1.153)	1.050	(0.736, 1.499)	1.018	(0.727, 1.424)
Snorting, gasping or stopping breathing, ≥1/week	1.048	(0.950, 1.155)	Ref	0.792	(0.507, 1.237)	1.053	(0.714, 1.553)	1.044	(0.710, 1.535)
Day sleepiness	1.030	(0.926, 1.144)	Ref	1.019	(0.695, 1.494)	1.441	(1.001, 2.075)	1.183	(0.824, 1.700)
Sleep problems	1.045	(0.969, 1.126)	Ref	1.051	(0.792, 1.394)	1.095	(0.838, 1.431)	1.415	(1.019, 1.965)
Obstructive sleep apnea	0.969	(0.893, 1.051)	Ref	0.812	(0.621, 1.062)	1.006	(0.755, 1.341)	0.846	(0.630, 1.136)

* Adjusted for sociodemographic variables (age, sex, race, economic status, marital status, and job status), health-related variables (body mass index, history of cardiovascular disease and cancer, hypertension, diabetes mellitus, and depressive symptom), and behavioral variables (physical activity, sedentary activity, caffeine consumption, alcohol drinking, and cotinine). OR: Odds Ratios, CI: Confidence Intervals.

## Data Availability

The data used in this study are available from the National Health and Nutrition Examination Survey (NHANES) database.

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
