# Peer review of "Effect of Ethylene Oxide Exposure on Sleep Health: Using NHANES Data from 2015 to 2020"

_healthcare, 2024, doi:10.3390/healthcare12242499_

Round 1
Reviewer 1 Report
Comments and Suggestions for Authors
Thank you for the opportunity to review this manuscript. The manuscript “Effect of ethylene oxide exposure on sleep health: Using NHANES data from 2015 to 2020” presents a study that investigates the effects of ethylene oxide exposure on sleep duration and sleep quality. Considering the broader context of presented studies and further impact of the published papers, I have some suggestions in the introduction, results and discussion section of the manuscript suggested to be addressed before the next steps in publication.
1. There were differences among study participants for age. As, age is one of the important factors that affect sleep duration and quality. I would suggest performing stratified analysis for age. Furthermore, the results could be discussed as comparative analysis in line with the previous literature in the discussion section of the manuscript.
2. Although the authors have well controlled the analysis for various confounding factors, I would suggest performing stratified/sensitivity analyses for the statistically significant parameters i.e., marital status, job/business etc. Also, as per literature, these parameters have significant contributions in sleep duration and quality.
3. I would suggest including some details about sleep duration and quality patterns for normal individuals and discussing their associations with the health impacts.
Minor comments:
1. I would suggest replacing “Ethylene oxide” in key words with suitable similar word. As it has already been used in the title of the manuscript.
Author Response
Below, we provide a detailed explanation of how each comment was addressed, along with our responses.
Comment 1 : There were differences among study participants for age. As, age is one of the important factors that affect sleep duration and quality. I would suggest performing stratified analysis for age. Furthermore, the results could be discussed as comparative analysis in line with the previous literature in the discussion section of the manuscript.
Response 2 : Thank you for your valuable feedback. As per your suggestion, we conducted a stratified analysis to examine the effect of ethylene oxide exposure on sleep health by age groups. The results have been included in "Supplementary Table 2," and the description of the findings has been updated in Results and Discussion, and Abstract section accordingly.
Comment 2 : Although the authors have well controlled the analysis for various confounding factors, I would suggest performing stratified/sensitivity analyses for the statistically significant parameters i.e., marital status, job/business etc. Also, as per literature, these parameters have significant contributions in sleep duration and quality.
Response 2 : Thank you for your valuable feedback. As you suggested, we conducted a stratified analysis to examine the effect of ethylene oxide exposure on sleep health based on marital status and employment type, which were significant control variables in the univariate chi-square analysis. The results have been included in "Supplementary Table 3" and "Supplementary Table 4," and the findings have been incorporated into Results and Discussion, and Abstract section of the manuscript.
Comment 3 : I would suggest including some details about sleep duration and quality patterns for normal individuals and discussing their associations with the health impacts.
Response 3 : Thank you for your valuable feedback. As per your advice, we have summarized previous studies on the factors related to sleep duration and sleep quality that affect the health of individual populations. Additionally, we have emphasized the necessity of improving sleep health through a review of the previous research, and this content has been added to the Introduction section.
Comment 4 : I would suggest replacing “Ethylene oxide” in key words with suitable similar word. As it has already been used in the title of the manuscript.
Response 4 : Thank you for your valuable suggestion. We have replaced the keyword "ethylene oxide" in the Abstract with the broader term "Chemical Exposure."
All revisions in response to the comments have been marked in red in the manuscript.
Should further clarification or additional revisions be required, we would be happy to make the necessary adjustments.
Thank you for your time and consideration.

Reviewer 2 Report
Comments and Suggestions for Authors
Under the Methods section, consider providing more explanation for the Cronbach's alpha thresholds, as these are lower than commonly accepted standards. Cite sources to support this decision.
In the Discussion section, consider Providing actionable suggestions for how the SPQ could be adapted or supplemented in future studies to improve its reliability and validity. Please add how this study contributes uniquely to the field, especially in areas where previous studies may have fallen short.
In addition, please add clear subheadings to enhance readability in both the Results and Discussion sections to enhance the readability (e.g., "Practical Implications," "Future Research Directions," and "Strengths and Limitations"). Also, pay more attention to the formatting (e.g., tables), and please use more recent studies as your references.
Author Response
Comment 1 : Under the Methods section, consider providing more explanation for the Cronbach's alpha thresholds, as these are lower than commonly accepted standards. Cite sources to support this decision.
Response 1 : Thank you for your valuable comment regarding the lower-than-commonly-accepted Cronbach's alpha thresholds in our Methods section. In response, we have provided a detailed explanation to justify these thresholds in the context of our study. Specifically, we have clarified how the exploratory nature of the research and the limited number of items influenced the alpha values and cited relevant literature to support this approach. The revised explanation has been incorporated into the Methods section.
Comment 2: In the Discussion section, consider Providing actionable suggestions for how the SPQ could be adapted or supplemented in future studies to improve its reliability and validity. Please add how this study contributes uniquely to the field, especially in areas where previous studies may have fallen short.
Response 2: Thank you for your thoughtful suggestion. In response, we have provided actionable recommendations in the Discussion section for enhancing the reliability and validity of the Sleep Pattern Questionnaire (SPQ) in future studies. These include integrating objective sleep data from wearable devices and incorporating additional variables such as daytime activity levels and stress assessments. These revisions are detailed in the updated into Discussion section of manuscript.
Comment 3: In addition, please add clear subheadings to enhance readability in both the Results and Discussion sections to enhance the readability (e.g., "Practical Implications," "Future Research Directions," and "Strengths and Limitations"). Also, pay more attention to the formatting (e.g., tables), and please use more recent studies as your references.
Response 3: Thank you for your valuable feedback. As per your suggestion, we have added subheadings to the Results and Discussion sections to improve readability. Additionally, we have adjusted the formatting of the tables (especially, logistic regression tables) to align with publication standards. In the Discussion section, we have incorporated recent studies to enhance the reflection on our research methods and findings.
Should further clarification or additional revisions be required, we would be happy to make the necessary adjustments.
Thank you for your time and consideration.

Round 2
Reviewer 1 Report
Comments and Suggestions for Authors
Thank you for the opportunity to review the revised version of the manuscript entitled “Effect of ethylene oxide exposure on sleep health: Using NHANES data from 2015 to 2020”. Based on comments/suggestions and their implementations by the authors, the manuscript has improved manifolds. I have no further specific comments. I wish the best of luck with the next steps in the publication.